# The association of preoperative psychological assessments with long-term sleeve gastrectomy surgery outcomes

Sofia Yeremian[1], Ara Keshishian[2]*, Robert Mardirosian[3]

1 Kirk Kerkorian School of Medicine at UNLV, Las Vegas, Nevada, United States of America,
2 Department of General Surgery, Huntington Hospital, Pasadena, California, United States of America,
3 Department of Mathematics, Glendale Community College, Glendale, California, United States of America

☉ These authors contributed equally to this work.
* ara.keshishian@med.usc.edu

## Abstract

### Background

Nearly all third-party healthcare plans adhere to the 1991 National Institutes of Health guidelines, which require preoperative psychological assessments before bariatric surgery. Despite these guidelines, the impact of these assessments on the long-term weight loss outcomes in laparoscopic sleeve gastrectomy patients without pre-existing psychological disorders remains ambiguous and insufficiently explored.

### Objective

The objective of this study is to identify any relationship between preoperative psychological assessments and the long-term weight loss outcomes of patients having undergone a laparoscopic sleeve gastrectomy procedure without a diagnosed psychological disorder at the time of surgery.

### Methods

Data from the Metabolic and Bariatric Surgery Accreditation and Quality Improvement Program registry for 120 patients who underwent laparoscopic sleeve gastrectomies (2008–2022) were retrospectively reviewed. The average percent weight loss was compared between those with (n = 61) and without (n = 59) preoperative psychological evaluations. All patients had given their consent for data collection for research purposes.

### Results

Statistical comparisons using a two-tailed t-test revealed no statistically significant difference in the average percent weight loss between patients who underwent and

**Data availability statement:** All relevant data are available from the BioStudies repository (accession number S-BSST2861) at the following link: https://www.ebi.ac.uk/biostudies/studies/S-BSST2861.

**Funding:** The author(s) received no specific funding for this work.

**Competing interests:** NO authors have competing interests Enter: The authors have declared that no competing interests exist.

did not undergo psychological evaluations (27.76% vs. 29.42%, p = 0.37) before a laparoscopic sleeve gastrectomy.

## Conclusions

In this retrospective cohort, no statistically significant association was observed between preoperative psychological evaluation and long-term weight loss outcomes. This raises questions about the necessity of these assessments, suggesting they may act as unnecessary barriers to care. Further research is required to determine whether preoperative psychological evaluations should be specifically aimed at patients with pre-existing mental health conditions instead of being applied universally to all candidates for bariatric surgery.

## Introduction

Obesity is a significant public health concern, impacting more than one-third of American adults [1], with a reported prevalence of obesity at 41.5% and severe obesity at 20.7% [2]. With obesity on the rise since 1999, projections indicate it could reach 55.6% in men and 80.0% in women by 2030 [3].

Common lifestyle interventions like exercising or commercial diet programs to address obesity are ineffective in achieving sustained, long-term weight loss [4]. Likewise, pharmaceutical therapeutics for weight loss, such as GLP-1 agonists, have shown promising early results but also significant side effects [5]. The long-term outcome of stopping the GLP-1 is unknown [6]. Exhibit inadequate efficacy and unsustainable long-term excess weight loss; furthermore, they carry a broad range of short and long-term complications and side effects [7]. Endoscopic bariatric surgical procedures, including endoscopic gastric reduction techniques and balloons, have shown minimal sustained long-term weight loss, with adjustable gastric balloons demonstrating higher complication rates than laparoscopic bariatric surgery [8–10]. On the other hand, bariatric procedures, including duodenal switches (DS), along with their single anastomosis variant (SIPS/SADI), and sleeve gastrectomies have superior outcomes in long-term weight loss maintenance and improvements in obesity-related comorbidities compared to non-surgical methods [11–14].

Following the establishment of the 1991 National Institutes of Health (NIH) Bariatric Consensus Statement, the standard criteria for bariatric surgery require multidisciplinary evaluations by medical, surgical, nutritional, and psychiatric professionals to assess candidacy for bariatric surgery and enhance outcomes by helping patients understand the lifelong post-surgical changes [15]. While these NIH requirements have led to insurance companies mandating preoperative psychological evaluations before bariatric surgeries, literature on the role of these evaluations in predicting weight loss outcomes is mixed, short-term, and inconsistent [16]. Some studies suggest that factors like depression and aggression may contribute to weight regain following bariatric surgery [17–19], while others report no significant association between presurgical psychological conditions—depression, anxiety, history of sexual

abuse, self-esteem, and binge eating—and short-term weight loss outcomes [20,21]. The mixed results on this topic are further complicated by concerns about the construct validity of depression self-report measures, which may limit their reliability in predicting weight loss outcomes [22]. Additionally, weight gain as a side effect of mood stabilizers, antidepressants, and other psychotropic medications raises the question of whether postoperative weight gain is due to the underlying mental health condition or the treatment itself [23–26]. Furthermore, psychological evaluations often overlook critical physiological and socioeconomic factors, such as metabolic changes in hormones (like leptin and ghrelin), environmental influences, gender, preoperative BMI, and genetics [27–29].

Along with the inconsistent relationship between psychological evaluations and bariatric surgery outcomes [18,19] across various studies and the overlooking of additional, significant weight-altering factors [17], many studies also generalize their findings across different bariatric procedures or focus specifically on the Roux-en-Y gastric bypass [22]. The laparoscopic sleeve gastrectomy has become increasingly preferred due to its superior weight loss results, better resolution of comorbidities, fewer complications, and shorter operating time compared to other surgeries [30–32]; however, despite its growing popularity, there is a notable lack of research individually examining the impact of psychological evaluations on weight loss success following the laparoscopic sleeve gastrectomy, leaving a noteworthy gap in the literature. This study aims to clarify the role of presurgical psychological evaluations in predicting long-term weight loss outcomes, specifically for patients undergoing laparoscopic sleeve gastrectomy, addressing current inconsistencies in the literature and considering both psychological and physiological factors to enhance the effectiveness of bariatric surgery.

The published literature in support of the psychological evaluation in support of pre-operation workup for weight loss surgery is riddled with opinions, surveys, assumptions, and professional prejudices and biases, and no scientific data in support of this requirement other than in those cases where there is a pre-existing mental health concern [18,33]. The requirement for preoperative psychological evaluation is a form of prejudice [34], a barrier to access, and a threat to health equity.

## Materials and methods

This study is a retrospective review of the Metabolic and Bariatric Surgery Accreditation and Quality Improvement Program (MBSAQIP) and Surgical Review Corporation (SRC) data registries, as well as the subject charts of 120 patients, spanning the period from 2008 to 2022. These patients had sought a consultation at SRC and MBSAQIP-certified bariatric surgical practices. Data collected from 2008 to 2012 was initially stored in the SRC database before being integrated into the MBSAQIP's database in 2012. Subsequently, this unified database encompassed previously stored and newly collected data from 2012 to 2021. The laparoscopic sleeve gastrectomy (LSG) procedures analyzed in this study were performed at a teaching hospital or an outpatient surgical center based on the patient's medical history, surgical risks, the procedure performed, and third-party reimbursement requirements. All patients had given their consent for the use of their data for research purposes.

Data were gathered regarding the patients' demographics, procedure type, date of surgery, pre- and postoperative comorbidities, and weight measurements. The data was anonymized, and all identifiable information, including name and patient ID numbers, was removed. Due to the retrospective data analysis conducted on previously collected data and the anonymized status of the information, no Institutional Review Board (IRB) approval was necessary. Patients with missing data were contacted via email and phone to ensure a complete dataset and to avoid selection bias. The patients were then divided into two categories: those who underwent a psychological evaluation and those who did not.

Those who did not disclose any history of psychological or mental illness or relevant medications during both their bariatric surgery and primary care consultations, and whose insurance did not extend coverage to bariatric procedures, were exempt from undergoing psychological evaluation and clearance before surgery. Regardless of their insurance coverage, patients who had previously received mental health care or were on anxiolytic or psychotropic medication were required to have a psychological evaluation by a psychiatrist or psychologist of their choosing as a part of their presurgical workup.

The patients' psychological conditions were chronic and were managed with medication under the supervision of a psychologist or psychiatrist. For this subset of patients, clearance for the LSG was only approved upon receipt of a psychological evaluation clearance by their healthcare provider.

Based on the collected data, metrics such as pre- and postoperative body mass indices (BMIs) and percentage weight loss were calculated. These continuous variables were presented as means with standard deviations and ranges. Statistical comparisons between continuous variables were conducted using a two-sample t-test. The analysis was performed using StatPlus software (StatPlus: Mac, AnalystSoft Inc., Version 8. https://www.analystsoft.com/en/).

### Ethics statement

The dataset used in this study was fully de-identified prior to analysis, with all direct patient identifiers, including names, dates of birth, and medical record numbers, removed to protect patient confidentiality. The data were derived from the Metabolic and Bariatric Surgery Accreditation and Quality Improvement Program (MBSAQIP) registry and institutional clinical records and contain sensitive health information governed by third-party data use agreements; therefore, unrestricted public sharing is not permitted. No additional ethical or legal restrictions were imposed by an Institutional Review Board, as IRB approval was not required for analysis of anonymized data. Access to the de-identified dataset may be granted upon reasonable request to the corresponding author: Ara Keshishian, MD, MPH, FACS, FASMBS (email: ara.keshishian@med.usc.edu, who will coordinate approval with the data-holding institution or governing registry.

## Results

### Laparoscopic sleeve gastrectomy patient demographics

The study comprised 120 participants, 87 females (72.5%) and 33 males (27.5%). This gender disproportion mirrors findings from recent studies in the field, which indicate that approximately eighty percent of patients undergoing bariatric or metabolic surgery identify as female [35].

Among the 120 participants, 61 of those individuals (50.8%) underwent a psychiatric evaluation prior to the procedure. The mean age of these 61 participants was 42.7 years (SD = 10.75), with an age range of 21.43 to 7and30 years. On average, the time elapsed since the operation for these 61 patients was 5.89 years (SD = 3.02), ranging from 2.08 to 16.33 years. The average preoperative BMI for this cohort was 39.32 kg/m² (SD = 5.62), ranging from 21.74 kg/m² to 54.92 kg/m². On average, it took 2.36 years (SD = 1.97) to reach the lowest weight, with a range of 0.42 to 13.58 years. The average, lowest postoperative BMI within this cohort was 25.17 kg/m² (SD = 4.53), ranging from 18.17 kg/m² to 42.6 kg/m². The mean percent weight loss was 27.76% (SD = 10.28%), ranging from 3.38% to 54.07%. The demographic makeup of this LSG cohort with psychiatric evaluations is outlined in Table 1.

Among the 120 LSG patients, 59 (49.2%) were not subjected to a psychological evaluation. As outlined in Table 1, the average time elapsed since this subgroup's operation was 3.47 years (SD 0.91), with a range of 2.00 to 6.42 years. The average age among this subgroup was 41.78 years (SD = 10.59), ranging from 20.04 to 61.55 years. Prior to the operation, the average BMI was 38.82 kg/m² (SD = 7.26), ranging from 26.62 kg/m² to 59.13 kg/m². On average, these individuals experienced a period of 2.52 years (SSD = 3.92 until they reached their lowest weight, with a range of 0.58 to 31.42 years. The average lowest postoperative BMI recorded was 24.99 kg/m² (SD = 4.06), ranging from 16.30 kg/m² to 41.97 kg/m². Notably, the mean weight loss percentage was 29.42% (SD = 10.05%), with individual reductions ranging from 0.59% to 57.98%.

### Weight loss comparison for laparoscopic sleeve gastrectomy patients with and without psychological evaluation

Of the 120 patients who underwent the LSG procedure, 50.8% (n = 61) underwent a psychological evaluation, while 49.2% (n = 59) did not. As indicated in Table 2, in the cohort of LSG patients with psychological evaluations, the mean percentage

**Table 1. Demographic makeup for LSG patient cohorts.**

| Procedure | Psychological evaluation? (Yes/No) | Number of patients | Gender distribution | Average age (years) | Average time since operation (years) | Average preoperative BMI (kg/m2) | Average time of lowest weight (years) | Average lowest BMI (kg/m2) | Average weight loss (%) |
|---|---|---|---|---|---|---|---|---|---|
| Laparoscopic Sleeve Gastrectomy | Yes | 13 | Female:7 (53.8%) Male:6 (46.2%) | 48.76 years (SD = 15.07) | 11.23 years (SD = 4.32) | 41.45 kg/m² (SD = 8.83) | 3.21 years (SD = 3.46) | 24.57 kg/m² (SD = 2.75) | 32.58% (SD = 14.49%) |
| | No | 58 | Female:49 (84.5%) Male:9 (15.5%) | 41.16 years (SD = 10.47) | 3.46 years (SD = 0.87) | 38.73 kg/m² (SD = 7.29) | 2.50 years (SD = 3.95) | 24.96 kg/m2 (SEM = 4.09) | 29.34% (SD = 10.12%) |

**Caption**: This table compares laparoscopic sleeve gastrectomy outcomes for patients who did and did not undergo a psychological evaluation before surgery. Both groups achieved similar average lowest BMIs (24.57 kg/m² for those with evaluations and 24.96 kg/m² for those without) and comparable average weight loss percentages (32.58% and 29.34%, respectively). The data indicate that psychological evaluations do not influence weight loss outcomes, as both groups show similar results.

**Table 2. Two-tailed t-test comparison of the average percent weight loss in LSG patient cohorts with and without psychological evaluations.**

**Descriptive statistics**

| VAR | N | Mean | Std Dev | Variance | Minimum | Maximum |
|---|---|---|---|---|---|---|
| % weight loss-SLV-Psych (1) | 13 | 0.33 | 0.14 | 0.02 | 0.03 | 0.54 |
| % weight loss = SLV-No-Psych (2) | 58 | 0.29 | 0.10 | 0.01 | 0.01 | 0.58 |
| Means Report | | | | | | |
| VAR | Mean | 95% LCL | 95% UCL | | | |
| % weight loss-SLV-Psych (1) | 0.33 | 0.24 | 0.41 | | | |
| % weight loss = SLV-No-Psych (2) | 0.29 | 0.27 | 0.32 | | | |
| Mean Difference (1–2) | 0.03 | −0.04 | 0.10 | | | |
| t-test assuming equal variances (homoscedastic) | | | | | | |
| Hypothesized Mean Difference | 0.00 | | | | | |
| Mean Difference | 0.03 | | | | | |
| Pooled Variance | 0.01 | | | | | |
| Test Statistic | 0.96 | | | | | |
| Degrees of Freedom | 69 | | | | | |
| H1: Mu1 – Mu2 ≠ 0 / Not equal (two-tailed) | | | | | | |
| t Critical Value (5%) | 1.99 | p-value | 0.34 | H1 (5%) | Rejected | |
| H1: Mu1 – Mu2 < 0 / Less than (lower-tailed) | | | | | | |
| t Critical Value (5%) | −1.67 | p-value | 0.83 | H1 (5%) | Rejected | |
| H1: Mu1 – Mu2 > 0 / Greater than (upper-tailed) | | | | | | |
| t Critical Value (5%) | 1.67 | p-value | 0.17 | H1 (5%) | Rejected | |

**Caption**: This table presents the results of a two-tailed t-test comparing the average percent weight loss between laparoscopic sleeve gastrectomy (LSG) patients who underwent a psychological evaluation and those who did not. Descriptive statistics are provided, including means, standard deviations, and ranges for each cohort. The t-test results indicate no significant difference in weight loss between the two groups, with a p-value of 0.34, suggesting that psychological evaluations do not significantly impact weight loss outcomes in LSG patients.

of weight loss from the highest preoperative weight to the current weight was 27.76% (SD = 10.28%), while for those without a psychological evaluation, the mean weight loss was 29.42% (SD = 10.05%). According to a two-tailed t-test, the null hypothesis (H0) posited that there was no difference in the mean percent weight loss between the two groups (μ1 = μ2),

while the alternative hypothesis (H1) suggested that there was a difference ($\mu1 \neq \mu2$). The calculated p-value was 0.37. The null hypothesis was not rejected, as the p-value is more significant than the $\propto$ $\alpha$level of 0.05. Therefore, there was no statistically significant difference in the mean percent weight loss between the LSG patients who did and did not undergo a preoperative psychological evaluation.

## Discussion

### Implications of the study

Although various research studies have explored the impact of psychological evaluations on long-term weight loss across different bariatric surgery procedures, this is the first objective scientific study to assess the efficacy and practicality of mandated preoperative psychological assessment on the long-term outcomes of solely laparoscopic sleeve gastrectomy procedures. Within the parameters examined, comparing weight loss outcomes among patients undergoing LSG with and without preoperative psychological evaluations showed no statistically significant differences in the stable average percent weight loss. As laparoscopic sleeve gastrectomy emerges as a more refined option for surgical weight loss, this study highlights the critical need to reassess the role and efficacy of mandatory preoperative psychological evaluations in bariatric surgery. Additionally, it calls for further investigation into alternative factors that may impact weight loss outcomes in bariatric surgery patients.

Furthermore, while psychological evaluations are intended to assess readiness for surgery and optimize weight loss outcomes following a bariatric procedure, they also serve as an additional burden, contributing to access inequality in bariatric surgery [36,37]. Factors such as cost, availability of mental health resources, long wait times for appointments, and subjective interpretations of evaluation results could influence access to bariatric surgery and subsequent weight loss interventions [37]. Therefore, future research should explore the impact of psychological evaluations on weight loss outcomes and investigate how these evaluations may affect equitable access to bariatric care for diverse patient populations.

Beyond predicting weight loss, preoperative psychological evaluation in bariatric surgery serves important ethical and preventive functions. These assessments aim to ensure that patients understand the irreversible nature of surgery, are prepared for lifelong behavioral changes, and can adapt to the psychosocial consequences of substantial weight loss [38,39]. Psychological evaluation may also help identify patients at risk for poor postoperative adjustment or decisional regret, supporting informed consent and patient safety. Although no association with weight loss outcomes was observed in this study, these broader clinical considerations remain relevant, particularly for patients with known or suspected mental health vulnerabilities.

### Limitations of the study

The limitations of this study include the relatively small sample size and the difference in the average time elapsed since surgery between the two patient cohorts, which may influence observed weight loss trajectories. In addition, potential selection bias should be acknowledged, as the grouping criterion of receipt of a preoperative psychological evaluation was not randomly assigned and overlapped with the presence or absence of disclosed psychiatric history and insurance requirements. Consequently, the comparison groups may differ in unmeasured clinical, psychosocial, or socioeconomic characteristics beyond the psychological evaluation itself, limiting causal inference. Although efforts were made to exclude patients with documented psychiatric diagnoses from the primary analytic question, residual confounding cannot be fully excluded. As such, the findings of this study should be interpreted as hypothesis generating rather than conclusive. Future prospective, longitudinal studies with larger patient populations and standardized psychological assessment criteria are needed to more definitively determine whether preoperative psychological evaluations independently influence long term weight loss outcomes following laparoscopic sleeve gastrectomy.

## Conclusions

In this retrospective cohort, no statistically significant association was observed between preoperative psychological evaluation and intermediate weight loss outcomes following laparoscopic sleeve gastrectomy. Although the study was not powered to exclude a clinically meaningful difference, these findings suggest that routine, non-selective preoperative psychological evaluation may have limited predictive value for postoperative weight loss. Importantly, the absence of an observed association in this dataset does not imply that psychological assessment lacks clinical relevance, but rather that its role may require refinement and more targeted application rather than universal implementation. Further prospective studies with larger sample sizes are warranted to clarify whether psychological evaluations restricted to individuals with pre-existing mental health diagnoses or identified psychosocial vulnerabilities independently influence long-term weight loss outcomes, or whether broader requirements for evaluation contribute to barriers in accessing bariatric care.

## Author contributions

**Conceptualization:** Ara Keshishian.

**Data curation:** Ara Keshishian, Sofia Yeremian, Robert Mardirosian.

**Formal analysis:** Ara Keshishian, Sofia Yeremian, Robert Mardirosian.

**Investigation:** Ara Keshishian, Sofia Yeremian.

**Methodology:** Ara Keshishian.

**Software:** Robert Mardirosian.

**Visualization:** Sofia Yeremian, Robert Mardirosian.

**Writing – original draft:** Sofia Yeremian.

**Writing – review & editing:** Ara Keshishian, Sofia Yeremian.

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
