## [Decision Letter · Decision Letter 0]

10 Dec 2025

Dear Dr. Keshishian,

Thank you for submitting your manuscript to PLOS ONE. After careful consideration, we feel that it has merit but does not fully meet PLOS ONE’s publication criteria as it currently stands. Therefore, we invite you to submit a revised version of the manuscript that addresses the points raised during the review process.

We look forward to receiving your revised manuscript.

Kind regards,

Kwang-Sig Lee

Academic Editor

PLOS One

**Journal Requirements:**

Reviewers' comments:

Reviewer's Responses to Questions

**Comments to the Author**

1. Is the manuscript technically sound, and do the data support the conclusions?

Reviewer #1: Yes

2. Has the statistical analysis been performed appropriately and rigorously?

Reviewer #1: Yes

3. Have the authors made all data underlying the findings in their manuscript fully available?

Reviewer #1: Yes

4. Is the manuscript presented in an intelligible fashion and written in standard English?

Reviewer #1: Yes

Reviewer #1: Overall, this is an interesting and well-structured paper on a clinically relevant issue.

The topic deserves attention, and the dataset is analyzed clearly.

To strengthen the manuscript and make it more balanced, I suggest the following minor revisions:

1. Study Design and Bias

- The grouping criterion (having or not having a psychological evaluation) overlaps strongly with the presence or absence of psychiatric history, which introduces potential selection bias.

- Please acknowledge this explicitly in the Limitations section and clarify that the study's findings are hypothesis-generating rather than conclusive.

2. Interpretation of "No Significant Difference"

- The statement that "no significant difference was found" should not be interpreted as evidence that psychological evaluations are unnecessary.

- Consider rephrasing to: "In this retrospective cohort, no statistically significant association was observed between preoperative psychological evaluation and long-term weight loss outcomes. However, the study was not powered to exclude a clinically meaningful difference."

3. Clinical and Ethical Role of Psychological Evaluation

- Beyond predicting weight loss, psychological assessment in bariatric surgery serves an ethical and preventive function — ensuring that patients understand the irreversible nature of surgery, are psychologically ready for lifestyle changes, and minimizing postoperative regret or maladaptation.

- A short paragraph reflecting this broader purpose would make the Discussion more comprehensive and clinically grounded.

4. Conclusion Section

- Consider tempering the conclusion's tone. For example: "While no significant difference was observed in this dataset, the role of psychological evaluation may need refinement and better targeting rather than elimination."

- This phrasing preserves the originality of your message while avoiding overgeneralization.

.

Reviewer #1: No

---

## [Author Response · Author response to Decision Letter 1]

7 Jan 2026

Response to Reviewers

Manuscript ID: PONE-D-25-37468

Title: The association of preoperative psychological assessments with long-term sleeve gastrectomy surgery outcomes

Dear Academic Editor and Reviewers,

We thank the Academic Editor and Reviewer #1 for their thoughtful and constructive comments, which have helped improve the clarity, balance, and methodological transparency of our manuscript. We have carefully addressed each point raised and revised the manuscript accordingly. Our point-by-point responses are provided below.

Academic Editor Comments

Data Availability and Ethics Statement

Comment:

PLOS ONE requires clarification regarding ethical or legal restrictions on data sharing, including who imposed them and how data may be accessed. The ethics statement should appear only in the Methods section.

Response:

We have revised the Methods section to include a dedicated subsection describing data governance, de-identification procedures, and access restrictions. We clarify that the dataset was fully de-identified before analysis, with all direct patient identifiers removed, and that the data are governed by third-party data use agreements (MBSAQIP/SRC), which preclude unrestricted public sharing. We further specify that an Institutional Review Board imposed no additional ethical or legal restrictions, as IRB approval was not required for analysis of anonymized data. Access to the de-identified dataset is available upon reasonable request through the corresponding author. The ethics statement now appears exclusively within the “Methods” section, in accordance with journal requirements.

Reviewer #1 Comments

1. Study Design and Bias

Comment:

The grouping criterion overlaps with psychiatric history, introducing potential selection bias. This should be acknowledged in the “Limitations,” and the findings should be framed as hypothesis-generating.

Response:

We acknowledged this potential source of selection bias in the “Limitations” section. We now clarify that the non-random assignment to psychological evaluation groups overlaps with psychiatric history and insurance requirements, which may introduce residual confounding and limit causal inference.

2. Interpretation of “No Significant Difference”

Comment:

The absence of a statistically significant difference should not be interpreted as evidence that psychological evaluations are unnecessary. The reviewer suggests rephrasing to note that the study was not powered to exclude a clinically meaningful difference.

Response:

We have revised the language to adopt the suggested phrasing. The manuscript now states that no statistically significant association was observed between preoperative psychological evaluation and weight loss outcomes in this retrospective cohort, while explicitly acknowledging that the study was not powered to exclude a clinically meaningful difference. This revision avoids overinterpretation while preserving the integrity of our findings.

3. Clinical and Ethical Role of Psychological Evaluation

Comment:

Psychological assessment serves ethical and preventive functions beyond predicting weight loss. A short paragraph reflecting this broader role should be added to the “Discussion.”

Response:

We have added a concise paragraph to the “Discussion” acknowledging the ethical and preventive role of psychological evaluation in bariatric surgery, including assessment of patient readiness, understanding of the irreversible nature of surgery, and capacity to adapt to long-term behavioral and psychosocial changes. This addition clarifies that our findings relate specifically to weight loss outcomes and do not negate the broader clinical relevance of psychological assessment, particularly in patients with known or suspected mental health vulnerabilities.

4. Conclusion Section

Comment:

The conclusion should be tempered to suggest refinement and better targeting of psychological evaluation rather than elimination.

Response:

We have revised the “Conclusion” accordingly. The updated language emphasizes that, while no significant association was observed in this dataset, the role of psychological evaluation may warrant refinement and more targeted application rather than universal implementation. This phrasing preserves the originality of our findings while avoiding overgeneralization.

Additional Revisions

• We reviewed the manuscript for consistency with PLOS ONE style requirements and corrected minor formatting issues.

• References were reviewed for accuracy and completeness; no retracted articles were cited.

• All revisions are highlighted in the marked-up manuscript submitted with track changes.

We appreciate the opportunity to revise our manuscript and believe that these changes have strengthened its methodological rigor and clinical balance. We thank the Academic Editor and Reviewer for their valuable feedback and hope the revised manuscript is now suitable for publication in PLOS ONE.

Sincerely,

Ara Keshishian, MD, MPH, FACS, FASMBS

(on behalf of all authors)

---

## [Editor Report · Decision Letter 1]

26 Feb 2026

The association of preoperative psychological assessments with long-term sleeve gastrectomy surgery outcomes

PONE-D-25-37468R1

Dear Dr. Keshishian,

We’re pleased to inform you that your manuscript has been judged scientifically suitable for publication and will be formally accepted for publication once it meets all outstanding technical requirements.

Kind regards,

Kwang-Sig Lee

Academic Editor

PLOS One
---

## [Editor Report · Acceptance letter]

PONE-D-25-37468R1

PLOS One

Dear Dr. Keshishian,

I'm pleased to inform you that your manuscript has been deemed suitable for publication in PLOS One. Congratulations! Your manuscript is now being handed over to our production team.

Kind regards,

on behalf of

Professor Kwang-Sig Lee

Academic Editor

PLOS One